# A Review of the Biosynthesis and Structural Implications of Insulin Gene Mutations Linked to Human Disease

**DOI:** 10.3390/cells12071008

**Published:** 2023-03-25

**Authors:** Sara Ataie-Ashtiani, Briony Forbes

**Affiliations:** Flinders Health and Medical Research Institute, College of Medicine and Public Health, Flinders University, Adelaide, SA 5042, Australia

**Keywords:** insulin gene mutations, neonatal diabetes, mutant insulin, insulin biosynthesis

## Abstract

The discovery of the insulin hormone over 100 years ago, and its subsequent therapeutic application, marked a key landmark in the history of medicine and medical research. The many roles insulin plays in cell metabolism and growth have been revealed by extensive investigations into the structure and function of insulin, the insulin tyrosine kinase receptor (IR), as well as the signalling cascades, which occur upon insulin binding to the IR. In this review, the insulin gene mutations identified as causing disease and the structural implications of these mutations will be discussed. Over 100 studies were evaluated by one reviewing author, and over 70 insulin gene mutations were identified. Mutations may impair insulin gene transcription and translation, preproinsulin trafficking and proinsulin sorting, or insulin-IR interactions. A better understanding of insulin gene mutations and the resultant pathophysiology can give essential insight into the molecular mechanisms underlying impaired insulin biosynthesis and insulin-IR interaction.

## 1. Introduction

During the past 50 years, investigations into the structure and function of insulin, the insulin tyrosine kinase receptor (IR), as well as the signalling cascades, which occur upon insulin binding, have revealed the many roles insulin and its signalling plays in cell metabolism and growth [1,2]. It has become clear that insulin signalling is not only fundamental in tissues previously identified as “insulin sensitive” (such as adipose, liver, and muscle tissue), but in almost all tissues of the body [3,4]. Accordingly, precise regulation of insulin biosynthesis and its role in signalling is crucial for health [2]. Nonetheless, despite the disease impact, much remains unknown about the molecular mechanisms underlying impaired insulin biosynthesis [5] and insulin-IR interactions [6].

It is estimated that 1 in 11 adults has diabetes, with 12% of health expenditure, globally, being diabetes-related [7]. Diabetes mellitus is a collection of metabolic diseases characterised by chronic hyperglycaemia due to diminished insulin secretion and synthesis and/or increased insulin resistance [8]. Recent technological advances have enabled molecular exploration of diabetes occurring in childhood and non-obese patients, thus shedding light on single gene defects causing such diseases [5]. Monogenic diabetes counts for 1–5% of all diabetes cases and can result from mutations in genes encoding β-cell potassium channels, β-cell glucose-sensor glucokinase, transcription factors, and insulin [9]. 

Insulin gene mutations can cause the development of unique diabetes subtypes [10]. Such mutations can impair the folding of insulin, cause the produced mutant to be retained in the endoplasmic reticulum (ER), trigger ER stress and β-cell death, as well as impair insulin-IR interaction [11]. Through one or more of these mechanisms, insulin gene mutations can cause a variety of conditions. These include neonatal-onset diabetes mellitus (NDM), diagnosed before the age of 12 months, Mutant Insulin-gene Induced Diabetes of Youth (MIDY), often emerging during childhood, and Maturity Onset Diabetes of the Young (MODY), often emerging during adolescence [12,13]. Two types of monogenic neonatal diabetes also exist: transient (TNDM), which spontaneously remits, and permanent (PNDM), requiring lasting treatment. It is important to note that TNDM may relapse later in life [14]. Furthermore, most MIDY and MODY cases are caused by heterozygous mutations, and pathology can involve mutants exerting a dominant negative impact on the expression of the wild-type allele [15]. 

Monogenic diabetes is often misdiagnosed and mistreated as type 1 or type 2 diabetes. Consequently, incorporating genetic testing and an understanding of disease-causing mutations into clinical care can be highly beneficial for patient experience and health outcomes [16]. Furthermore, the exploration of the structural changes in insulin, resulting from genetic mutations, and the molecular biology involved can supply crucial information for the development of novel and improved insulin analogues used to treat diabetes [2,3,17]. Recently, multiple excellent reviews have identified and collated disease-causing insulin gene mutations and discussed their clinical impacts [4,8,11]. Here, we will not only assemble the results of the literature, noting insulin gene mutations as a cause of disease, but also give a unique insight into the possible structural implications of these mutations. Through the integration of underlying structural biology and the resultant clinical expression of the discussed mutations, gaps in knowledge can be accurately identified and a stage for insulin analogue design is provided [6].

## 2. Insulin Expression—From Genome to Protein

Insulin biosynthesis occurs in β-cells found in the islets of Langerhans within the pancreas. The human insulin gene contains three exons separated by two introns [18]. As seen in Figure 1, only exons 2 and 3 contribute to the coding of mature insulin. Exon 1 contains the 5′ untranslated region and has a regulatory role in insulin expression. Exon 2 encodes the signal peptide, B-chain, as well as part of the connecting peptide (C-peptide). Finally, exon 3 encodes the rest of the C-peptide and the A-chain [4,8]. Preproinsulin, the single-chain precursor to mature insulin, consists of four domains: the amino-terminal signal peptide, the B-chain, the C-peptide, and the carboxy-terminal A-chain. Mature insulin consists of two independent peptide chains: the A-chain (21 residues) and the B-chain (30 residues), which are connected by two disulfide bonds between cysteine residues at A7–B7 and A20–B19. Additionally, an intra-chain disulfide bond is present within the A-chain connecting residues A6–A11. These three disulfide bonds stabilise the core structure comprising the three α-helices of insulin [19]. Mutations hindering the formation of these bonds prevent the proper folding of the mature protein and often have detrimental clinical consequences [8,20].

During insulin biosynthesis (Figure 2), the insulin gene is transcribed to mRNA with the aid of various transcription factors. Subsequently, the translation of preproinsulin occurs at the cytosolic surface of the ER and the protein is translocated into the ER. Within the ER, the signal peptide is cleaved off by signal peptidase to form proinsulin. Proinsulin, in turn, folds at the luminal side of the ER, forming the three evolutionarily conserved disulfide bonds fundamental to insulin’s structure [20]. At this stage, misfolding of mutant proinsulin can initiate the unfolded protein response (UPR) and lead to ER stress and β-cell dysfunction over time [21]. Wild-type proinsulin exits the ER and moves to the Golgi body where it is sorted and placed into secretory vesicles. The C-peptide is endoproteolytically excised by prohormone convertase 1/3 and 2 (PC1/3 and PC2) and carboxypeptidase E (CPE) within post-Golgi vesicles producing mature insulin. The hormone is then stored in the secretory vesicles of pancreatic β-cells as hexamers stabilised by zinc [22]. Upon a secretory signal, mature insulin is released into the bloodstream. Certain mutations can prevent proper sorting, leading to mutants undergoing constitutive exocytosis rather than regulated exocytosis. In this case, mutants are continually secreted regardless of extracellular factors or secretory signals [23]. Once secreted, insulin binds to IR and triggers a series of structural changes, leading to IR intracellular phosphorylation events that provide docking sites for signalling molecules that results in the initiation of signalling cascades [2]. This life cycle of insulin biosynthesis is illustrated in Figure 2. 

The IR forms an (α-β)_2_ disulfide bonded heterodimer composed of two αβ monomers. The α-subunits are linked by intermonomer disulfide bonds. The two monomers come together with an ectodomain containing several subdomains (L1, CR, L2, FnIII(1–3), and αCT) and an intracellular region. The extracellular α-subunits and the N-terminus of the β-subunits contain the ligand binding regions. A transmembrane helix of the β-subunit spans the membrane. Cytoplasmic domains (JM, TKD, C-tail) are the target of insulin-dependent phosphorylation during signalling. Two main insulin signalling pathways are activated upon insulin-IR interaction. These include the phosphatidylinositol 3-kinase (PI3K)/AKT pathway, which primarily stimulates metabolism and glucose uptake, and the mitogen-activated protein kinase (MAPK) pathway, which principally stimulates mitogenic activity and growth signalling [24]. Through these two pathways, insulin and the related insulin-like growth factors play an essential role in the control of metabolism, as well as growth and development [2,25]. 

## 3. Methods and Search Criteria

This review was conducted by searching the National Centre for Biotechnology Information (NCBI) PubMed (January 2022), using terms including “insulin gene mutations”, “INS-gene mutations”, “MODY”, “MIDY”, “mutant insulin”, and “insulin biosynthesis”. Case studies and reviews investigating mutations in the human insulin gene as a cause of or in association with diseases, such as diabetes, MODY, MIDY, NDM, hyperglycaemia, or growth disorders, published after 2000, were analysed. The primary sources cited in the selected literature were also investigated. Studies involving both adults and children were considered. In vitro and in silico investigations of the identified mutant insulins were also reviewed to enable the discussion of the molecular mechanisms of pathogenesis and structural implications of the identified mutations. The results from the selected studies were obtained and evaluated by one reviewing author (S. A-A). 

## 4. Insulin Gene Mutations

Tager et al. were the first to describe a mutant human insulin in 1979, and over 70 insulin gene mutations have been identified in the years which have followed [8,26]. Here, we endeavour to give insight into the effects these insulin gene mutations have on insulin biosynthesis and insulin-IR interaction. These mutations are categorised by the stage of the insulin life cycle they most notably interfere with:Transcription and translation.Preproinsulin trafficking and proinsulin sorting.Insulin-IR interaction.

Most of the identified mutations thus far are heterozygous. Additionally, not all deleterious mutations will result in disease presentation in early life, as the expression of a wild-type allele may compensate for the mutation present. However, subsequent heterozygous mutations can lead to an increased risk of developing insulin-linked diseases if the expressed mutant impedes the secretion of wild-type insulin [11]. Additionally, factors including individual pancreatic β-cell dysfunction and β-cell turnover influence the likelihood and severity of disease in such cases.

### 4.1. Mutations Affecting Transcription and Translation

Within this subsection, mutations that impact the effectiveness of transcription and translation, or those resulting in frameshifts, premature stopping of translation (truncation), and altered mRNA stability, are discussed. Such mutations often result in disease onset during infancy, minimal circulating insulin, and are homozygous (unless indicated otherwise in the section below) [8,27]. Figure 3 highlights the location of these mutations along the insulin gene. 

Deleterious promoter mutations lead to reduced gene expression by preventing gene activation and mRNA production by disrupting the recruitment of transcription factors. Three such single-point substitutions, **c.-331C>G**, **c.-331C>A**, and **c.-332C>G**, impair promoter activity and lead to NDM [27,28,29]. Interestingly, a transcriptional activity reduction of up to 90% has been observed in pancreatic β-cell lines due to these mutations, while **c.-339G>A** was not observed to significantly alter transcriptional activity [27]. These results and the clinical presentation of these mutations highlight the possible importance of the two cytosines at sites c.-331 and c.-332 for the regulation of insulin gene expression. Correspondingly, **c.-332_-331CC>GG** is a two-base substitution at this location, resulting in TNDM, and **c.-331del** is a single-base deletion at this site, resulting in NDM [4]. Conversely, **c.-218A>C** results in NDM (both permanent and transient types observed) as it likely disrupts the cyclic adenosine monophosphate response element 3 (CRE3) site within the insulin gene promoter region, which interacts with numerous DNA binding proteins, resulting in reduced insulin expression [27,30]. Lastly, **c.-366_-343del** is a heterozygous 24-base deletion of the evolutionarily conserved C1 and E1 elements of the insulin gene promoter (binding sites for MafA and Neuron Differentiation 1 transcription factors) and results in PNDM, due to substantially reduced insulin expression [27,31]. These findings highlight the importance of the CRE3, cytosines at c.-331 and c.-332, and C1 and E1 sites for insulin expression in humans. 

Mutations impeding mRNA stability, processing, and function can result in reduced gene expression and pathology. For example, **c.*59A>G** is a mutation within the 3′ untranslated section of the insulin gene that reduces mRNA stability, thus decreasing post-transcriptional gene expression and resulting in NDM [27,32]. To elaborate, investigations in a heterozygous lymphoblastoid cell line have highlighted low levels of **c.*59A>G** mRNA transcripts compared to wild-type transcripts, likely due to this mutation impeding mRNA polyadenylation [27]. Alternatively, **c.187+241G>A** is believed to interfere with appropriate splicing by resulting in the favoured detection of a new splice site. In this case, NDM is likely caused by the production of two mutant transcripts predicted to undergo decay [33]. Newly identified insulin gene mutations **c.187+2T>C** and **c.187+5G>C**, in patients with NDM, likely have a similar method of pathogenesis, although further research is required for better understanding [27]. A heterozygous substitution mutation resulting in MODY, **c.188-31G>A**, located within intron 2, is predicted to create a novel splice site and cause the insertion of 29 bases from intron 2 in the exon [34,35]. Mutations **c.188-37T>A**, **c.188-40C>A**, **c.188-1G>A**, and **c.188-15G>A** cause NDM and likely pose issues with correct splicing and lead to the production of mutant transcripts [4]. The duplication mutation **c.212dupG** results in MODY by causing a frameshift and the production of a non-functional protein [36]. Finally, **c.1A>G**, **c.2T>C**, **c.3G>T**, and **c.3G>A** are single-point substitutions at the start of exon 2 [4,27]. These mutations impact the start codon and result in NDM by altering the preproinsulin translation initiation site [27,37]. Real-time PCR of insulin mRNA abundance revealed the same mRNA levels for **c.3G>T**, **c.3G>A**, and wild-type insulin. However, HeLa cells transfected with **c.3G>A** and **c.3G>T** demonstrated 86% and 79% reduction in insulin content, respectively, compared to cells transfected with the wild-type insulin gene. This supports the notion that these mutations impede insulin biosynthesis during translation [27].

Deletion mutations can result in frameshifts. For example, homozygous **c.-65_581del** (646-base deletion) causes PNDM, whereas it increases the risk of the development of diabetes in adulthood, when present as a heterozygous mutation [27,38]. **c.-370-?_186+?del** (557-base deletion) removes a segment of the insulin gene promoter, exon 1, and exon 2, and results in PNDM as mature insulin cannot be expressed. Similarly, **p.M1_N110del** and **p.M1_Q62del** are large deletions spanning exons 2 and 3, leading to the inability to express functional insulin and NDM [4].

Nonsense mutations are single-base substitutions that produce a stop codon and result in truncated proteins due to the premature termination of translation [39]. The **p.W4*** (signal peptide mutation) and **p.R46*;**(**B22**) mutations are predicted to result in PNDM by preventing the expression of full-length insulin [4]. Although **p.R46*;**(**B22**) has an intact signal peptide, it is unable to undergo ER translocation, implicating this previously unrecognised region as performing key regulatory functions in insulin trafficking. This is an area requiring further research [40]. The **p.Q62*** (**c.184C>T**) is a point mutation located at the end of exon 2, and **p.Q78*** (**c.233del**) is a heterozygous single base deletion. Both of these mutations result in a protein truncated within the C-peptide, which lacks the A-chain and causes PNDM and MODY, respectively [4,8,11,27,34]. Likewise, heterozygous **p.Y108*** is an A-chain mutation that results in a premature stop codon and a truncated protein, leading to NDM. This mutant exhibits partial ER retention, partial recruitment to post-Golgi vesicles, and a concomitant reduction in the secretion of co-expressed wild-type insulin [11,41]. Lastly, heterozygous **p.Y103*** results in a protein truncated at the A-chain and is a cause of diabetes diagnosed after the age of 10 (MODY), possibly due to the accumulation of ER stress due to mutant misfolding and β-cell dysfunction over time [4]. 

### 4.2. Mutations Affecting Preproinsulin Trafficking and Proinsulin Processing

Mutations in this category are predominantly heterozygous and can impact the final peptide in one of four ways. Firstly, mutations introducing a new cysteine residue, or those that alter an existing cysteine residue, will generate an unpaired cysteine, which can lead to aggregation and ER stress. Secondly, mutations of residues neighbouring a cysteine may impact and hinder disulfide bond formation. Thirdly, mutations that introduce a change in an α-helix, or impact a structurally important residue, will cause instability and limited insulin folding. Fourthly, mutations may impact the correct processing or trafficking of the produced mutant. Additionally, mutations in residues important for receptor interaction may alter insulin bioactivity, even if the mutant is successfully secreted (discussed in Section 4.3). Figure 4 highlights the location of such mutations impacting trafficking, processing, and IR interactions on the amino acid sequence of preproinsulin.

#### 4.2.1. Signal Peptide Mutations

Signal peptide mutations can impair trafficking across the ER, lead to mutant accumulation in the ER, the UPR, and ultimately cause β-cell dysfunction [8]. If the signal peptide’s secondary structure is altered, it can no longer effectively serve the function of directing protein trafficking [42]. The mutations **p.R6C** and **p.R6H** (**c.17G>A**) lower insulin secretion by impacting a conserved region of the signal peptide, which facilitates protein alignment in the ER throughout protein translocation [43,44]. Interestingly, the **p.R6H** mutant is, in part, targeted normally to secretory granules, yet still results in ER stress and mild MODY onset after the age of 10. Moreover, cell studies have highlighted that cells transfected with **p.R6C** demonstrate substantially compromised proinsulin release, while **p.R6H** secretion does not significantly vary from wild-type secretion [45]. This observation may be explained as the introduction of an additional cysteine may result in misfolding and/or aggregation. The **p.P9R** (**c.26C>G**) and **p.L14R** are signal peptide mutations cited as a cause of NDM through β-cell dysfunction, likely due to the introduction of a large and positively-charged arginine residue, in place of a comparatively smaller proline, that confers conformational rigidity and a hydrophobic non-polar leucine residue, respectively [4,46]. Although **p.A2T** does not affect preproinsulin translocation, it impairs signal peptide cleavage and thus, prevents the production of proinsulin, resulting in MODY type 10 [47]. Heterozygous **p.A23S** has been observed to cause NDM, although further research into the pathophysiology of this mutation is required [48]. Similarly, **p.A24V** likely inhibits signal peptide cleavage, causing a lack of insulin secretion and PNDM [4,49]. The **p.A24D** mutation results in the production of two subpopulations of peptides that undergo impaired folding and are retained in the ER, resulting in PNDM [41,50]. **p.A22P** is cited as a cause of NDM, although further investigation into the method of pathogenesis is required [4]. Finally, **p.L13R** (**c.38T>G**) has been observed to cause the congenital absence of β-cells and PNDM. This is possibly due to severe ER stress caused by the altering of the highly conserved L13 residue, which is key in the formation of the hydrophobic core of the signal peptide [42].

#### 4.2.2. C-Peptide Mutations

Although not part of mature insulin, the C-peptide plays an essential role during insulin biosynthesis by facilitating correct folding and disulfide bond formation between the A and B chains [51]. For example, p.L68M leads to young-onset type 2 diabetes and may cause the production of a non-functional mutant [44]. Additionally, p.G84R is a mutation linked to NDM that is predicted to disrupt the folding of proinsulin, resulting in ER stress and β-cell apoptosis [11]. The p.R89H (proinsulin Tokyo), p.R89P (proinsulin Oxford), and p.R89L (proinsulin Kyoto) are processing site mutations at the C-peptide to A-chain junction, which cause issues with C-peptide cleaving, leading to hyperproinsulinemia and resulting in NDM [11,41,52,53,54]. Additionally, p.R89C is a cysteine mutation that causes PNDM through the addition of a cysteine residue at the cleavage site between the A-chain and C-peptide. Mutations at this residue disturb proinsulin processing; hence, patients exhibit high levels of circulating insulin with only the B-chain to C-peptide junction being cleaved [11,54].

#### 4.2.3. Mutations of Cysteine Residues

Mutations at cysteine residues result in insulins with an odd number of thiol groups and the disturbance of the formation of the three disulfide bonds integral to insulin’s structure [55]. Folding within the ER requires the formation of the 31–96, 43–109, and 95–100 (B7–A7, B19–A20, and A6–A11) disulfide bridges to stabilise the three α-helices of insulin and its receptor binding surfaces [19]. Therefore, the disulfide bonds of wild-type insulin form both interior supportive struts (at B19–A20 and A6–A11) and inter-chain staples (B7–A7, B19–A20) [56,57]. Their roles have been defined by extensive folding studies and chemically synthesised insulin analogues in which cysteine pairs have been substituted with serine [58,59,60,61]. When expressed in mammalian cell lines, human insulin cysteine mutants are often not secreted and are degraded by the UPR machinery [62].

The **p.C43G;**(**B19**)*,*
**p.C43S;**(**B19**)*,*
**p.C43Y;**(**B19**)*,* and **p.C43F;**(**B19**) are heterozygous cysteine mutations within the B-chain linked to PNDM. Mutations at this residue result in the inability to form one of two disulfide bonds between the B-chain and A-chain [4,41,63,64]. Additionally, **p.C109F;**(**A20**)*,*
**p.C109Y;**(**A20**)*,*
**p.C109R;**(**A20**)*,*
**p.C109G;**(**A20**)*,* and **p.C109S;**(**A20**) also cause NDM by preventing the formation of the A20–B19 disulfide bridge [4,65,66,67]. Similarly, **p.C31Y;**(**B7**) and **p.C31R;**(**B7**) result in NDM by hindering appropriate folding and leading to ER retention and stress, due to preventing the formation of the B7–A7 disulfide bridge [4,52]. **p.C31_L35del;**(**B7_B11**) involves the deletion of the B7 cysteine residue [4]. **p.C96S;**(**A7**) and **p.C96R;**(**A7**) are A-chain cysteine mutations, linked to PNDM, and result in complete mutant retention in the ER [44,52,53,54]. **p.C96Y;**(**A7**) is a MIDY-causing mutation at the same residue. Computer modelling suggests that this mutant partially folds into an intermediate, with a flexible N-terminus, which can cause the recruitment of other proinsulin molecules to the unpaired cysteine at B7, thus leading to aggregation [41,55]. The **p.C95Y;**(**A6**)*,*
**p.C95W;**(**A6**)*,*
**p.C95F;**(**A6**)*,* and **p.C95R;**(**A6**) mutations result in NDM through attenuation of the secretion of co-expressed wild-type insulin, possibly due to promoting aggregation [4,11,41,68]. Furthermore, **p.C100W;**(**A11**)*,*
**p.C100G;**(**A11**)*,* and **p.C100Y;**(**A11**) result in NDM by preventing the formation of the A6–A11 disulfide bond and causing misfolding [4]. Finally, **p.I99_C100insSI;**(**A10_A11**) is a deletion of two amino acids, including the A11 residue, hence, resulting in NDM [54]. These results demonstrate the importance of the formation of the three discussed disulfide bonds and that mutations impacting residues involved in disulfide bond formation can cause proteotoxicity by promoting aberrant intermolecular cross-linking [55].

#### 4.2.4. Mutations Introducing Cysteine Residues

Mutations which introduce a new cysteine residue are similarly detrimental and cause misfolding by leading to abnormal inter or intra-molecule disulfide bonds. However, depending on the conformational folding of the resultant mutants, differences in ER stress severity and age of disease onset are observed [55]. **p.F48C;**(**B24**) is a B-chain mutation that introduces a cysteine residue, and the mutant is completely retained in the ER, leading to NDM [41,64,67]. **p.Y50C;**(**B26**), **p.G69C**(**C-peptide**)*,*
**p.G75C**(**C-peptide**), **p.S85C**(**C-peptide**)*,* and **p.S98C;**(**A9**) also cause NDM, likely due to impairing correct disulfide bond formation, leading to considerable misfolding [4,67,69]. Similarly, **p.S101C;**(**A12**), **p.Y103C;**(**A14**)*,* and **p.Y108C;**(**A19**) are cysteine mutations, resulting in NDM, as the produced mutants are retained in the ER and not efficiently secreted [11,44,70]. **p.R55C**(**C-peptide**) is a heterozygous cysteine mutation linked to MODY, ketoacidosis, and insulinoma-associated antigen-2 antibody-negative diabetes. This substitution involves the first of the two arginine residues at the site of proteolytic processing between the B-chain and the C-peptide, likely preventing appropriate cleavage of the C-peptide [11,71]. Similarly, **p.G90C;**(**A1**) is a heterozygous cysteine mutation impacting the A-chain to C-peptide cleavage site, and is cited as a cause of PNDM [11,41]. 

#### 4.2.5. Non-Cysteine Mutations Impacting Disulfide Bonding and Structural Integrity

Non-cysteine insulin gene mutations have clinical presentations varying in severity. It must be noted that even wild-type proinsulin exhibits limited folding efficiency, and molecular and cell-based approaches suggest that proinsulin has evolved at the edge of non-foldability [55,72]. **p.Q28del;**(**B4**) and **p.Q28_H29del;**(**B4_B5**) are NDM-causing deletions, hypothesised to interfere with disulfide bond formation [4]. Notably, **p.H29D;**(**B5**)*,*
**p.H29Q;**(**B5**)*,* and **p.H29Y;**(**B5**) alter a histidine residue containing an imidazole ring, which packs close to the A7-B7 disulfide bond in wild-type insulin [4,44,64]. As this histidine normally packs close to the A7–B7 disulfide bond and forms hydrogen bonds to carbonyl oxygens on the A chain, these mutations likely interfere with correct disulfide bond formation [73]. Thus, these mutations most likely disrupt the folding of proinsulin, decreasing the thermodynamic stability of the protein and leading to ER stress [11,44]. 

Within wild-type insulin, the leucine at residue B6 inserts into a cavity surrounded by the non-polar side chains of leucines at B11, B15, and A16 [55]. Consequently, mutations at this residue can be predicted to introduce structural perturbations. The **p.L30R;**(**B6**)*,* **p.L30P;**(**B6**)*,*
**p.L30V;**(**B6**)*,* and **p.L30Q;**(**B6**) are mutations linked to NDM, due to the impact on packing through the introduction of charged or polar side-chains into the non-polar cavity, leading to retention of the mutants in the ER [11,45,74]. The **p.L30M;**(**B6**) is better tolerated as the introduction of a linear non-polar residue is likely to have less impact on the local environment. Due to this, **p.L30M;**(**B6**) exhibits 50% ER retention and is associated with mild diabetes (MODY) [45]. 

The **p.G32S;**(**B8**)*,*
**p.G32C;**(**B8**)*,*
**p.G32V;**(**B8**)*,* and **p.G32R;**(**B8**) mutations have all been linked to PNDM, although variable ages of onset have also been observed [4,41,48,75,76,77]. These mutations result in partial ER retention, partial recruitment to granules, and the attenuation of the secretion of co-expressed wild-type insulin [48]. Notably, residue B8 plays an important function in disulfide bond formation as it neighbours the B7 cysteine [78].

**p.H34D;**(**B10**) (proinsulin Providence) prevents appropriate trafficking and sorting of proinsulin and leads to an increased amount of proinsulin sorting to constitutive granules. As these granules lack prohormone convertases, preproinsulin is secreted into the bloodstream, resulting in type 2 diabetes associated with familial hyperproinsulinemia [8,79]. The reason why **p.H34D;**(**B10**) proinsulin is missorted to the constitutive granules is unclear. Interestingly, this mutant has approximately 5 times greater receptor binding affinity compared to wild-type insulin, as further discussed in Section 4.3 [80]. **H34P;**(**B10**) is a mutation at this residue linked to PNDM, although further research into the method of pathogenesis is required [67]. Another mutation at this residue, **p.H34Y;**(**B10**)*,* has been cited as a cause of NDM [4].

The leucine at residue B11 is buried in wild-type insulin, is essential for correct formation of the B-chain α-helix, and contributes to the hydrophobic core of insulin [55,58]. Consequently, the mutations **p.L35P;**(**B11**)*,*
**p.L35Q;**(**B11**)*,* and **pL35M;**(**B11**), at this residue, likely disturb α-helical propensity, as well as structural stability, and may lead to misfolding, explaining their links to NDM [44,55,67,81]. 

The leucine at residue L39(B15) also packs near C43(B19) and F48(B24) in a shallow crevice between the B- and A-chains [55]. Thus, **p.L39P;**(**B15**) is predicted to introduce significant perturbations to the packing of the core. Additionally, **p.L39H;**(**B15**) inserts a polar aromatic side chain into the discussed crevice, and through this, destabilises this region. Furthermore, **p.L39V;**(**B15**) introduces a β-branched side chain and results in a comparatively subtle alteration to structure, presumably by reducing the propensity for the B-chain helix to form [4,55]. Nonetheless, all identified mutations at this residue have been cited as a cause of NDM [4]. Furthermore, **p.L39_Y40delinsH;**(**B15_B16**) is a B-chain mutation where a leucine and a tyrosine are replaced by a histidine, preventing appropriate folding within the ER [74]. 

Likewise, **p.L41P;**(**B17**) is a B-chain mutation causing NDM [4,82]. In wild-type insulin, this residue neighbours the cysteine at B19 within the α-helix at the central region of the B-chain [55]. Thus, it is likely that the introduction of proline in place of the hydrophobic side chain of leucine perturbs appropriate folding, leading to a non-functional protein. However, if some mutant was correctly folded and secreted, it is likely this mutation would impair insulin binding to IR site 2 (defined below in Section 4.3). Mutation to alanine at B18, **p.V42A;**(**B18**) (**c.125T>C**), is linked to diabetes diagnosed after the age of 10 (MODY) [83]. This mutation likely impairs the efficiency of core packing near the B19 cysteine, although only to a moderate degree as the introduction of an alanine likely maintains the α-helical structure. Additionally, **p.V42G;**(**B18**) (**c.125T>G**) results in NDM. Mutation to a smaller glycine is likely to enhance helix flexibility and produce a cavity in the core, lowering the efficiency of 43–109(B19–A20) disulfide formation [55]. Notably, this mutation severely impairs the secretion of insulin in vitro, indicating complete ER retention [84]. 

Within wild-type insulin, a β-turn is present at residues B20 to B23, after the central B-chain α-helix (B9-B19). This turn plays an important role in a key structural rearrangement required for receptor binding, but it is also required to position the C-terminal B-chain residues (B24–B30) against the B-chain helix in the unbound state [85]. This turn includes two evolutionarily conserved glycine residues at B20 and B23. **p.G44R;**(**B20**) (**c.130G>A**) is a heterozygous mutation that causes MODY. This mutant exhibits impaired oxidative folding in the ER, reduced ER export, ER stress, and resultant β-cell apoptosis [86]. Clearly, maintaining a small flexible glycine in this location, next to the B19 cysteine, is critical for folding and stability. Within the B20–B23 turn, residue B22 forms a hydrogen bond with the glutamate at A17, stabilising the insulin molecule [71]. **p.R46Q;**(**B22**) (**c.137G>A**) disrupts this critical interaction and leads to the development of MODY in patients [87]. The IR affinity of the **p.R46Q;**(**B22**) mutant was 57% of wild-type insulin. Interestingly, β-cells can secrete this mutant as it was found in patient serum and secreted by INS-1 cells in vitro. However, **p.R46Q;**(**B22**) mutant expression was associated with impaired β-cell function [87]. The effect of mutation at this conserved glycine is dramatic and disrupts disulfide formation. **p.G47V;**(**B23**) is retained in the ER, is partially recruited to granules, and is linked to PNDM [11]. **p.G47D;**(**B23**) is linked to diabetes diagnosed after the age of 10 [55]. Finally, **p.P52L;**(**B28**) impacts intermolecular disulfide bonds that not only involve the mutant, but also wild-type insulin, forming misfolded disulfide-linked proinsulin complexes, limiting the secretion of bioactive insulin and resulting in NDM [86]. 

Residues neighbouring the receptor-binding surface of insulin are of particular interest to analogue design as they offer an opportunity for enhanced activity and novel receptor contacts. Residue A8 is of interest, where alteration leads to increases in IR binding, with a threonine to histidine mutation (found in birds and fish) increasing insulin potency in humans [88]. The mutation **p.T97P;**(**A8**) has been linked to MIDY. Cell studies demonstrate elevated intracellular mutant proinsulin levels and dramatically impaired insulin secretion. Additionally, the expression of **p.T97P;**(**A8**) causes increased intracellular proinsulin aggregate formation and ER stress, due to the introduction of proline at this residue, which disrupts disulfide bond formation by neighbouring cysteines [4,89]. **p.T97S;**(**A8**) is another mutation at this residue which is likely to cause less significant issues in folding, as serine and threonine both possess polar uncharged side chains. Despite this, **p.T97S;**(**A8**) has been associated with NDM [4]. **p.S98I;**(**A9**) and **p.Q104R;**(**A15**) introduce isoleucine (hydrophobic) and arginine (positively charged) residues, respectively, in place of polar uncharged amino acids, leading to structural changes that have been linked to NDM [4,90].

Residue A16 has long been of interest in relation to insulin’s foldability and function, as leucine at this residue fills a potential cavity enclosed by the isoleucine at A2, tyrosine at A19, and leucine at B15, which are conserved nonpolar receptor-binding residues [72]. **p.L105P;**(**A16**) and **p.N107D;**(**A18**) are A-chain mutations within this region, leading to NDM [4,91]. The side chain of leucine at A16 is normally buried within the hydrophobic core of insulin in both free and receptor-bound states. Consequently, **p.L105P;**(**A16**) would likely introduce a destabilising cavity to this region, preventing appropriate oxidative folding. Although **p.L105V;**(**A16**) impairs both insulin chain combination and folding, once folding has been achieved, the resultant mutant is compatible with a wild-type-like crystal structure and has high biological activity, indicating that mutation of the conserved A16 leucine impacts folding efficiency, rather than receptor affinity [55,92]. **p.Y108D;**(**A19**)*,* and **p.Y108N;**(**A19**) exhibit partial ER retention and lead to neonatal-onset MIDY [4,41]. In wild-type insulin, tyrosine at A19 projects from a nonpolar crevice to expose its para-hydroxyl group. Thus, pathogenicity may result as these mutations lie in close proximity to the A20–B19 disulfide bond and may cause the mispairing of cysteines in this critical region [55].

### 4.3. Mutations Affecting Insulin-IR Binding

Not only do mutations affect preproinsulin and proinsulin trafficking and processing, but they can impact on insulin residues known to interact with the IR during binding, and this can impact on signalling. Insulin binds the IR through two surfaces originally defined by site-directed mutagenesis [93] and binding studies, and have been recently confirmed by structural studies [6]. The insulin receptor, comprising a (α-β)_2_ homodimer, also interacts with insulin through two regions: the high affinity binding site 1 and the low affinity site 2 [6,93]. The highly conserved receptor binding residues of insulin, termed the “classical binding surface” or “site 1”, include residues A1, A2, A3, A4, A19, and A21, as well as B-chain residues B8, B9, B11, B12, B16, B24, B25, and B26 [93]. This binding surface is involved in the “site 1” insulin-IR interaction [94] where these residues interact with the IR L1 domain of one monomer and the alpha C-terminal peptide from the opposite monomer (αCT’) (recently further defined as site 1a, Figure 5). Upon insulin binding, αCT’ is relocated across the surface of L1, and insulin residues B24–B26 rotate by approximately 50 degrees from the core of insulin, thus allowing the end of the B-chain to fold out from the insulin’s core to make room for the αCT’ segment. The opening of this molecular “hinge” (located at B23) facilitates the interaction of non-polar side chains from residues A2, A3, B12, B24, and B25 with IR [95]. Cryo-electron microscopy studies recently revealed that residues A8 and B10 lie in close proximity to the FnIII-1′ of the opposite monomer in the active conformation, and this secondary site 1 interaction has been defined as site 1b [17]. On Figure 5 site 1a and site 1b, residues that are impacted by mutation are highlighted. As discussed below, mutations involving site 1 residues can not only alter receptor interaction and lead to defective binding, but may also impair insulin clearance and alter its half-life [96]. This can have a range of biological implications as effective blood glucose control requires rapid insulin response and clearance [97]. 

**p.V92L;**(**A3**) (insulin Wakayama) is a mutation of the conserved valine at position A3 to leucine. This mutation impairs receptor binding by 500-fold, demonstrating the importance of this residue for receptor interaction and its subsequent activation [98]. Residue A3 makes direct contact with the *α*CT’ [6]. In addition, this mutation increases insulin half-life and results in MIDY [4,99]. **p.E93K;**(**A4**) is a mutation at the neighbouring A4 residue and has been linked to MODY, with diagnosis after the age of 10. Structural analysis suggests that the wild-type glutamic acid at this residue forms a hydrogen bond with IR *α*CT’ during binding. Thus, the substitution to lysine likely impairs this interaction. However, the clinical presentation in some individuals also suggests abnormal insulin processing [100]. **p.E93G;**(**A4**) and **p.E93V;**(**A4**) (**c.277G>A**) are mutations at this residue, linked to NDM, and are likely to impair αCT’ interaction by removing the negative charge offered by glutamic acid [4,100]. As seen in Figure 5, in the bound state, the side chain of tyrosine at residue A19 projects towards the N-terminus of the A-chain and lies close to αCT’. Consequently, the mutations **p.Y108D;**(**A19**) and **p.Y108N;**(**A19**), which introduce a negative charge and a polar uncharged side chain, respectively, if folded, are likely to impact the interaction with αCT’ and perturb high-affinity receptor binding [4,67]. 

A key interaction required for site 1 high affinity binding involves residue B24. Hence, mutations in this region can be detrimental to effective IR binding [95]. The side chain of phenylalanine at B24 acts as an anchor to site 1a by inserting into a hydrophobic patch formed between the L1 and αCT’ interface. Like several other mutations that affect residues involved in receptor binding, substitutions at B24 also invariably impact stability as the hydrophobic phenylalanine serves as a stabiliser of the core helical structure. **p.F48S;**(**B24**) (insulin Los Angeles) [6,101] and **p.F49L;**(**B25**) (insulin Chicago) lead to dramatically reduced receptor binding, and these mutant insulins have a prolonged half-life [8,26,102,103,104,105]. 

As mentioned above, **p.G32S;**(**B8**)*,*
**p.G32C;**(**B8**)*,*
**p.G32V;**(**B8**)*,* and **p.G32R;**(**B8**) have all been linked to PNDM, albeit also demonstrating variable ages of disease onset. These mutations occur at the conserved B8 glycine [4,41,48,75,76,77]. Whilst this B8 glycine lies in close proximity to binding site 1, these substitutions are unlikely to have a major impact on direct receptor interaction, but rather affect disulfide bond formation involving the neighbouring B7. Indeed, studies of synthetically produced proinsulin with substitutions at B8 highlight the importance of B8 in folding efficiency, secretion, and disulfide stability [106].

Recent cryo-electron microscopy studies of insulin, in complex with IR, reveal that A8 lies in close proximity to the FnIII-1′ domain at site 1b [17,107]. As previously discussed above, **p.T97P;**(**A8**) [89] and **p.T97S;**(**A8**) mutations at A8 impair insulin secretion and stability, most likely due to the close proximity to the A7–B7 disulfide bond [4,89]. Insulin analogue studies demonstrate that, if chemically synthesised, multiple substitutions can be tolerated at this position, but with some impact on receptor binding and signalling [108]. For example, analogue T97H;(A8) insulin has a 2-fold increased receptor binding affinity and T97F;(A8) has a 2-fold lower affinity, highlighting the role of residue A8 in directly contacting the receptor [88,109,110]. The synthesis and analysis of **p.T97P;**(**A8**) and **p.T97S;**(**A8**) insulin analogues, however, has not been reported and so it is not possible to comment on their impact on structural integrity and receptor binding affinities. Notably, a histidine substitution at this residue, observed in chicken insulin, has been suggested to be the cause of the approximately 5-fold increase in its human IR binding [111], also indicating the key role this residue plays in IR binding.

Similarly, the **p.H34D;**(**B10**) (Proinsulin Providence) and **p.H34Y;**(**B10**) mutants are associated with sorting and secretion defects, and yet a range of substitutions can be tolerated in recombinantly synthesised B10 analogues [112]. Indeed, **p.H34D;**(**B10**) insulin binds with higher affinity through interaction at site 1b and, interestingly, has increased mitogenic signalling potency [80].

## 5. Conclusions and Future Directions

Overall, recent advances in technology and investigations of insulin have revealed the array of essential roles this hormone plays in cell metabolism and growth [2]. Despite this, much remains to be explored regarding the method of pathogenesis for monogenic diabetes. In this review, the insulin gene mutations identified as a cause of disease, and the structural implications of these mutations, were discussed. Over 100 studies were evaluated by one reviewing author, and over 70 insulin gene mutations were identified. Notably, the detection of insulin gene mutations is likely to increase in the coming years due to the routine use of genetic testing in clinical care. Testing in combination with a better understanding of the molecular consequences of such mutations can help guide already established routine care to improve patient experience and reduce misdiagnosis. Despite this, the treatment for most patients with insulin gene mutations, resulting in disease (resulting in minimal to no bioactive insulin in circulation), will include the use of exogenous insulin. Further investigations into the structural consequences of the identified mutations will also be beneficial in this field, as the gathered information can provide a novel stage for insulin analogue design to improve patient outcomes and quality of life [17].

## Figures and Tables

**Figure 1 cells-12-01008-f001:**
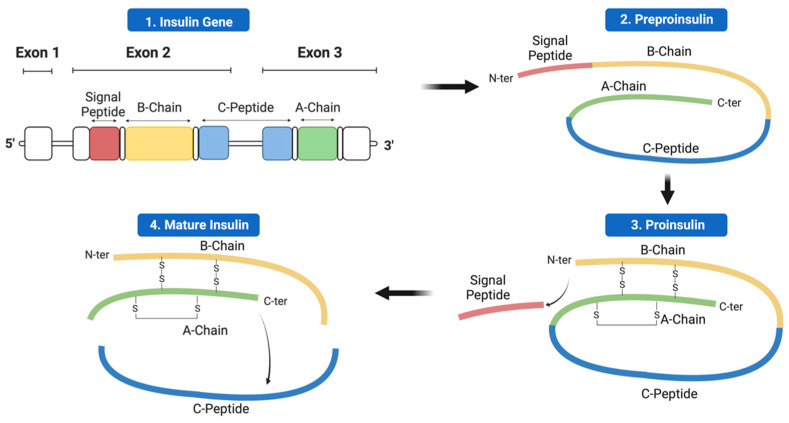
Schematic of the stages of insulin biosynthesis. 1. Insulin gene containing three exons separated by two introns. 2. Preproinsulin produced via the transcription and translation of the insulin gene. 3. Proinsulin produced from preproinsulin after the signal peptide is cleaved off by signal peptidase. 4. Mature insulin produced as the C-peptide is cleaved from proinsulin within secretory vesicles. Displayed domains: the amino-terminal (N-ter) signal peptide (red), the B-chain (yellow), the C-peptide (blue), and the carboxy-terminal (C-ter) A-chain (green).

**Figure 2 cells-12-01008-f002:**
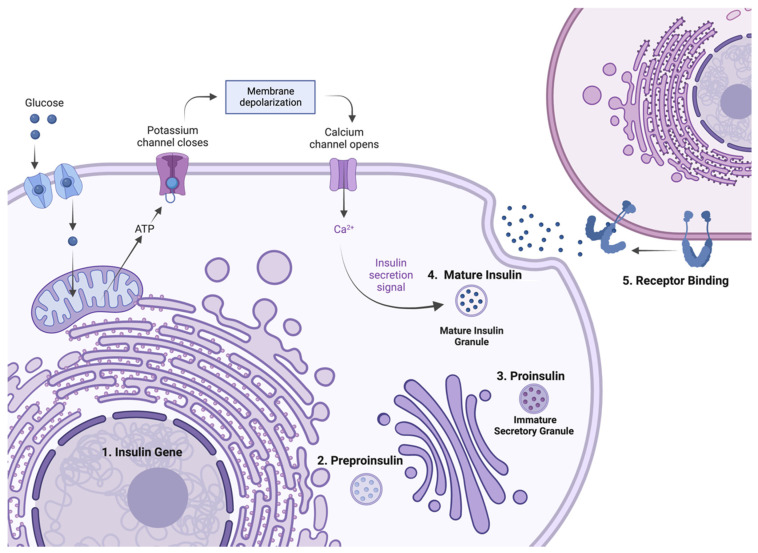
Schematic of the life cycle of insulin biosynthesis. After transcription within the nucleus (step 1), translation occurs at the cytosolic face of the Endoplasmic Reticulum (ER) (step 2), producing preproinsulin. Within the ER, the signal peptide is cleaved, disulfide bonds form, and the peptide is folded. Proinsulin moves into the Golgi body where it is sorted and packed into secretory vesicles (step 3). Within these vesicles, the C-peptide is cleaved off by prohormone convertases (PC1/3 and PC2) and carboxypeptidase E (CPE), forming mature insulin (step 4). Insulin is secreted in response to secretory signals that result in membrane depolarisation and calcium influx into the cell (top left). Insulin binds to IR (step 5) and the resultant receptor activation initiates intracellular signalling cascades.

**Figure 3 cells-12-01008-f003:**
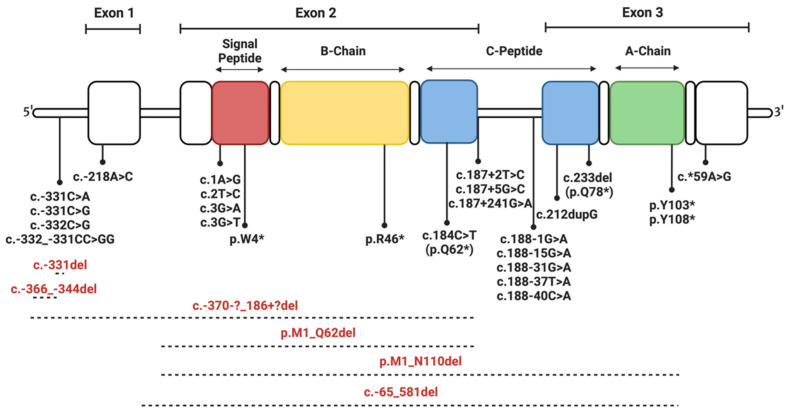
Schematic of the location of human insulin gene mutations affecting transcription and translation. Deletions are shown in red text and using dashed lines. Throughout this review, two naming systems are used. For mutations within the insulin gene, the “coding” system, as indicated by the presence of c. in front of the mutation name, is used. For mutations that affect the protein sequence, the “protein” naming system is used, as indicated by the presence of p. in front of the mutation name. Mutations resulting in protein truncations are annotated using the protein naming system, followed by “*”. Mutations starting with “–“ are located in the promoter or before the start of the coding sequence of the gene, while those starting with “*” are located after the coding sequence of the gene.

**Figure 4 cells-12-01008-f004:**
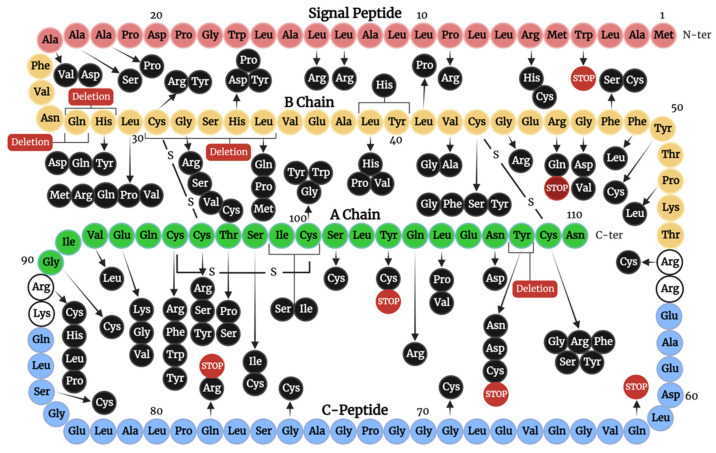
Schematic demonstrating the amino acid sequence of human preproinsulin with annotations of the location of insulin mutations impacting trafficking, processing, and IR interactions. Displayed domains: the amino-terminal (N-ter) signal peptide (red), B-chain (yellow), C-peptide (blue), and the carboxy-terminal (C-ter) A-chain (green). White residues are those which are cleaved during the conversion of proinsulin to insulin. Mutations resulting in stop codons and deletions are seen in red.

**Figure 5 cells-12-01008-f005:**
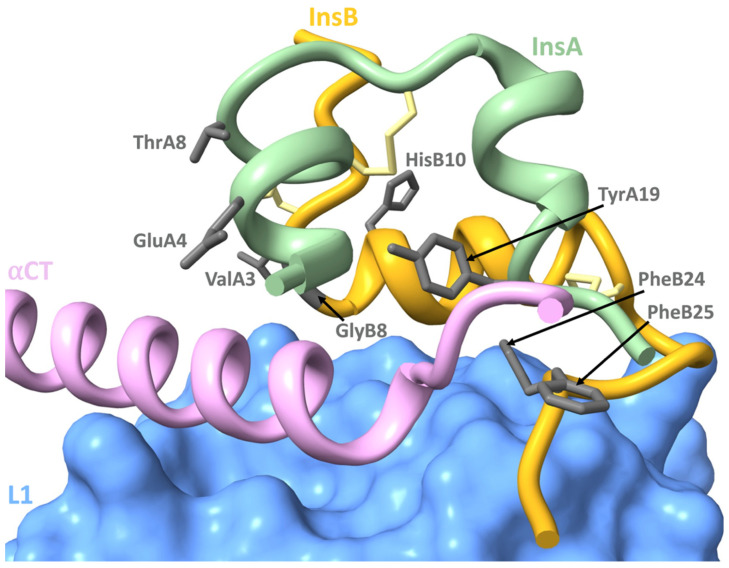
Schematic of wild-type insulin interacting with the insulin receptor (IR). Side chains of insulin residues key for IR interaction, with identified mutations, displayed in grey. Displayed domains: insulin B-chain (gold), insulin A-chain (light green), IR L1 (blue), IR αCT’ (pink). Insulin disulfide bonds are seen in yellow.

## Data Availability

Data sharing is not applicable to this article as no new data were created or analyzed in this review.

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
