# Peer review of "A Review of the Biosynthesis and Structural Implications of Insulin Gene Mutations Linked to Human Disease"

_cells, 2023, doi:10.3390/cells12071008_

Round 1

Reviewer 1 Report

This comprehensive review is the result of the evaluation of 100 studies of altogether 70 insulin gene mutations causing monogenic forms of diabetes (TNDM, PNDM, MIDY, MODY). Such mutations may impair insulin gene transcription and translation, preproinsulin trafficking and proinsulin sorting, or insulin-insulin receptor interactions. The study not only assembles the results of the literature evaluation but provides a unique insight into the underlying structural biology and the resultant clinical expression of the mutations.

The review is clearly written and systematic, and goes an important step beyond the usual reports of the clinical impact of individual mutations by providing mechanistic and structural interpretations of the mutations’ impact, and is likely to become a key reference in the field.

I have only minor comments and suggestions.

Line 405: proinsulin Providence: I would cite ref. 8.

Line 408: preproinsulin is secreted in the bloodstream: I would add “resulting in type 2 diabetes associated with familial hyperproinsulinemia”.

Lines 430 to 433: B17 mutation: it could be mentioned that such mutation would also impair insulin binding to the insulin receptor site 2 if the insulin was secreted.

Line 456: dramatic, not drammatic.

Line 480: reference 91 is the same as reference 72.

Line 499: ref. 6: the authors should also quote the upcoming comprehensive review article by Briony Forbes in Vitamins and Hormones.

Line 513: conformation, not conformaion.

Line 564: It could be added that substitutions at A8 explain the increased receptor affinity of chicken and turkey insulins and the decreased affinity of duck insulin, relative to human and porcine insulins.

Reviewer 2 Report

The discovery of the insulin hormone over 100 years ago and its subsequent therapeutic 8 application marked a key landmark in the history of medicine and medical research. The many roles insulin plays in cell metabolism and growth have been revealed by extensive investigations into the structure and function of insulin, the insulin tyrosine kinase receptor (IR), as well as the signaling cascades which occur upon insulin binding to the IR. In this review, the insulin gene mutations identified as causing disease and the structural implications of these mutations will be discussed.  Over 100 studies were evaluated by one reviewing author and over insulin gene mutations were identified. Mutations may impair insulin gene transcription and translation, pre-proinsulin trafficking and proinsulin sorting, or insulin-IR interactions.

A better understanding of insulin gene mutations and the resultant pathophysiology can give essential insight into the molecular mechanisms underlying impaired insulin biosynthesis and insulin-IR interaction.

The review is well-written and interesting and clearly describes insulin gene mutations identified so far. There are some minor issues to address to improve the paper.

1)      The abbreviation for the insulin gene is not mentioned in the text. I would add it in the Introduction.

2)      I would only expand just a bit the section on insulin mutations affecting receptor binding. What is the effect of these mutation on IR and/or downstream effectors activation levels? What is the biological relevance of insulin mutations prolonging insulin half-life?
